# The Association between Health Literacy and Gait Speed in Community-Dwelling Older Adults

**DOI:** 10.3390/healthcare8040369

**Published:** 2020-09-28

**Authors:** Kunihiko Anami, Shin Murata, Hideki Nakano, Koji Nonaka, Hiroaki Iwase, Kayoko Shiraiwa, Teppei Abiko, Akio Goda, Jun Horie

**Affiliations:** 1Department of Rehabilitation, Faculty of Health Sciences, Naragakuen University, Nara 631-8524, Japan; nonaka@naragakuen-u.jp; 2Department of Physical Therapy, Faculty of Health Sciences, Kyoto Tachibana University, Kyoto 607-8175, Japan; murata-s@tachibana-u.ac.jp (S.M.); nakano-h@tachibana-u.ac.jp (H.N.); shiraiwa@tachibana-u.ac.jp (K.S.); abiko@tachibana-u.ac.jp (T.A.); goda@tachibana-u.ac.jp (A.G.); horie-j@tachibana-u.ac.jp (J.H.); 3Department of Physical Therapy, Faculty of Rehabilitation, Kobe International University, Kobe 658-0032, Japan; iwase@kobe-kiu.ac.jp

**Keywords:** health literacy, gait speed, physical function, cognitive function, community-dwelling older adults

## Abstract

The association between gait speed, a vital health outcome in older adults, and health literacy, an important health promotion aspect, is unclear. This study examined the relationship of gait speed with health literacy, physical function, and cognitive function in community-dwelling older adults. The subjects were 240 older adults (52 men, mean age 73.8 ± 6.0 years). Gender, age, and education were self-reported, while height and weight were measured directly. Health literacy was evaluated using Communicative and Critical Health Literacy (CCHL). Grip strength, knee extension strength, toe-grip strength, sit-up test, sit-and-reach test, one-leg stance test time, 30-s chair-stand test (CS-30), and normal gait speed were measured. Subjects were divided into two groups based on normal gait speed—fast (speed ≥ 1.3 m/s) and slow (<1.3 m/s). In the logistic regression analysis, the dependent variable was normal gait speed (fast/slow). Four logistic regression models were utilized to determine whether health literacy affects gait speed. Height and CCHL were found to independently affect gait speed. That health literacy influences gait speed is a new discovery.

## 1. Introduction

As Japan faces a super-aged society never before experienced in human history, a variety of health outcomes are being evaluated both in communities and in hospitals by industrial, government, and academic sectors to understand the extent of disabilities and to enable disease prevention and long-term care prevention. Among these health outcomes, gait speed [1,2,3] has been identified as an extremely important indicator. Gait speed not only decreases with age [1,2,3,4] but affects both everyday activities [5] and quality of life [6,7,8]. Studenski et al. [3] explored gait speed and life expectancy in 34,485 community-dwelling adults aged 65 years or older with an average follow-up period of 12 years. They found that gait speed is correlated with survival rate; for each 0.1 m/s increase in gait speed, the risk of mortality drops by roughly 10%. A separate study by Montero-Odasso et al. [9] found that slower gait speed is associated with risks such as hospitalization. Gait speed has also been identified as a clinical indicator of the extent of disability in chronic respiratory [10,11,12] and cardiovascular disease [13].

The importance of gait speed is not limited to physical function. It has also been found to be correlated with cognitive function [14] and attention [15,16], which are essential for reading comprehension and have been linked to health literacy, a concept that has been drawing interest across the globe in recent years [17,18]. A systematic review by Sorensen et al. [19] summarizing existing definitions defined health literacy as “people’s knowledge, motivation, and competences to access, understand, appraise, and apply health information in order to make judgments and take decisions in everyday life concerning healthcare, disease prevention, and health promotion to maintain or improve quality of life during the life course.” A different systematic review by Berkman et al. [20] found that low health literacy was associated with increased hospitalizations, greater use of emergency care, and difficulty taking medications appropriately, and in older adults, it was associated with poorer overall health status and higher mortality rates. Wolf et al. [21] found that people with high health literacy engaged in more daily physical activity. The few studies that do exist have found that low health literacy is associated with low health status [22] and that health literacy is relatively low in Japan compared to European countries [23]. Uemura et al. [24] demonstrated that active learning for health promotion led to significant improvement in comprehensive health literacy and functional performance in community-dwelling older adults. Their results suggest that an active learning style and enhanced health literacy increased the amount of physical activity and dietary variety, which led to improved physical performance. The authors were unable to find research discussing the correlation between health literacy and gait speed in detail among the published literature on health literacy.

The number of studies on health literacy in Japan is limited [22,23,24,25,26]. Therefore, this study aimed to explore the relationship of gait speed with health literacy, physical function, and cognitive function in community-dwelling older adults.

## 2. Materials and Methods

### 2.1. Subjects

Subjects were 255 community-dwelling older adults signed up with a community older adult social exchange program who participated in the 2017 group physical fitness test. There were 54 men and 201 women with a mean age of 74.0 ± 6.2 years.

The exclusion criteria were as follows: (i) no warning signs of marked cognitive impairment, with a score of less than 24 on the Mini-Mental State Examination; (ii) in patients with cerebrovascular disease, Parkinson’s disease, and orthopedic diseases that impair gait; (iii) unable to complete all measurements. The final analysis used data from 240 subjects (Figure 1).

All subjects consented to participation before measurement after receiving both written and oral explanation of the research objective and a summary of the study. This study was carried out with approval from the Kyoto Tachibana University research ethics committee (approval no. 17–14).

### 2.2. Data Collection

#### 2.2.1. Basic Attributes

Subjects were asked to report their age, gender, education, and whether they had diseases (hypertension, dyslipidemia, diabetes, cardiovascular disease, respiratory disease, osteoporosis, and orthopedic disease) with a self-administered questionnaire. Height, weight, and body mass index were measured directly.

#### 2.2.2. Gait Speed

Gait speed was measured using a digital stopwatch on an 11-m walking path designed with a 5-m measurement section bordered on either side by 3-m preparation sections. At the start of measurement, subjects were verbally instructed to “Please walk normally” and their walking time was measured. Gait speed (m/s) was calculated using the time recorded (s).

#### 2.2.3. Health Literacy

Health literacy was assessed with the Communicative and Critical Health Literacy (CCHL), a self-administered questionnaire developed by Ishikawa et al. [25] to evaluate health literacy in members of the general public who do not have any notable diseases. The scale score is calculated using five items. A higher score indicates higher health literacy. Generally, health literacy consists of functional literacy, higher-order interactive literacy, and critical literacy. Functional literacy refers to the ability to read and write. Interactive literacy describes the development of skills that allow independent, knowledge-based action in a supportive environment. Critical literacy is the ability to critically analyze and use information to better control everyday events and circumstances. CCHL is comprised of interactive and critical literacies.

#### 2.2.4. Physical and Cognitive Function

Grip strength, knee extension strength, toe-grip strength, sit-up test, 30-s chair-stand test (CS-30), sit-and-reach test, and one-leg stance test time were measured as evaluations of physical function.

Grip strength was evaluated using a digital hand dynamometer (T.K.K. 5401 Grip-D, Takei Scientific Instruments, Niigata, Japan). This assessment tool has been shown to be valid and reliable [27]. Two measurements were taken for each side while standing upright with the arms hanging alongside the body and without allowing the dynamometer to touch the body. The maximum observed value (kg) was used.

Knee extension strength was measured using the method detailed by Bohannon [28]. This assessment tool has been shown to be valid [29] and reliable [30]. Knee extension strength was evaluated using a handheld dynamometer (μTasF-1, Anima Corporation, Tokyo, Japan). Measurements were taken with the subject seated in a chair with the knees flexed at 90° and the sensor pad fixed in place with a band at the distal end of the legs. Two measurements were taken for each leg. The body weight-normalized percentage (%), calculated by dividing the maximum observed value (kgf) by body weight (kg), was used.

Toe-grip strength was evaluated using a toe-grip dynamometer (T.K.K. 3362, Takei Scientific Instruments, Niigata, Japan). This assessment tool has been shown to be valid [31] and reliable [32]. Measurements were taken with the subject seated in a chair with the knees flexed at 90° and the ankles halfway between plantar flexion and dorsiflexion and internal and external rotation [33]. Two measurements were taken for each foot. The maximum observed value (kg) was used.

The sit-up test was measured using the method described by Abe et al. [34]. Subjects lay on a mat in a supine position with the knees bent at an angle of approximately 90°. The arms were crossed at the chest with the hands on opposite shoulders. Subjects performed a full sit-up to the upright position with their elbows touching their thighs and then returned to the supine position. The number of repetitions subjects could complete in 30 s was counted.

CS-30 was measured using the method described by Jones et al. [35]. Subjects began seated in a chair (height: 40 cm) with the arms crossed over the chest. They were instructed that reaching a standing position with the knees fully straightened and then sitting once again would be considered one cycle of movement and that, at the start signal, they should repeat that cycle as many times as possible for 30 s. The number of sit–stand–sit cycles completed in 30 s was counted.

The sit-and-reach test was evaluated using a digital sit-and-reach test box (T.K.K.5412, Takei Scientific Instruments, Niigata, Japan). This assessment tool has been shown to be valid [36]. The distance covered by the finger tips during the reach was measured. Two measurements were taken, and the maximum observed distance (cm) was used.

One-leg stance test time was evaluated using a digital stopwatch. The duration for which the subject could maintain a one-legged standing position with the eyes open was measured [37]. Two measurements were taken for each leg, and the maximum duration (s) was used. Time for the one-leg stance test was capped at 120 s.

Cognitive function was evaluated using the MMSE and Trail Making Test-A (TMT-A). The MMSE is widely used as a general test of cognitive function. It consists of 11 items yielding a maximum score of 30 points [38]. A score of 23 points or below is considered cognitive impairment [39,40]. TMT is a widely used test of attention [41]. We measured TMT-A for the purpose of evaluating selective attention and sustained attention. Subjects were asked to draw lines, as quickly as possible, connecting numbers 1 through 25, which were randomly arranged on a page, from smallest to largest. The time taken was recorded with a digital stopwatch.

### 2.3. Statistical Analysis

Subjects were first divided into groups by gait speed to conduct logistic regression analysis for factors related to fast and slow gait speed. Subjects were divided into groups as defined by Quach et al. [2]: one group with a gait speed of 1.3 m/s or above (fast) and another group below 1.3 m/s (slow). Statistical analysis was performed as follows.

We used independent-sample t-tests to examine continuous variables and chi-square tests to examine categorical variables between the fast and slow groups. Since the independent-sample t-test was performed for 15 factors, the *p*-value was set to 0.0033 by Bonferroni correction. Since the chi-square test was performed on eight factors, the *p*-value was set to 0.0063 by Bonferroni correction. φ and d values were calculated as measures of effect size. Next, logistic regression analysis was conducted with gait speed (fast or slow) as the dependent variable. Previous studies have revealed that gait speed is related to age, height, muscle strength, and attention [1,2,3,4,16]. Therefore, we created four logistic regression models to determine whether health literacy affects gait speed.
-Model 1. Independent variable was health literacy only.-Model 2. Independent variables were model 1 plus age, gender, and height.-Model 3. Independent variables were model 2 plus CS-30 and TMT-A.-Model 4. Independent variables were model 2 plus Knee extension strength and TMT-A.

Forward stepwise selection (likelihood ratio) was used to select variables. The significance standard for rejecting the null hypothesis was 5% for logistic regression analysis. Analyses were conducted using SPSS ver.25.

## 3. Results

After excluding subjects following the exclusion criteria, the data of 240 subjects were analyzed. Four subjects were excluded from analysis owing to medical history (e.g., cardiovascular disease), eight owing to MMSE score of 23 or below, and five owing to missing data. Final subjects were 52 men and 188 women with a mean age of 73.8 ± 6.0 years.

Of the 240 subjects, 200 were in the fast gait speed group (≥1.3 m/s) and 40 were in the slow group (<1.3 m/s). Height (*p* = 0.001, d = 0.560), CS-30 score (*p* < 0.001, d = 0.820), one-leg stance test time (*p* < 0.001, d = 0.670), and TMT-A score (*p* < 0.001, d = 0.720) were significantly higher in the fast group than in the slow group. Thus, the fast group had higher physical function and attention than the slow group (Table 1).

Table 2 shows the results of logistic regression analysis using walking speed as the dependent variable. In model 1, when the walking speed was analyzed using only the CCHL score as the independent variable, the OR was 1.222 (95% CI: 1.062–1.405), and the correct classification rate was 82.9%. For model 2, CCHL scores (OR: 1.231, 95% CI: 1.064–1.425) and height (OR: 1.091, 95% CI: 1.033–1.152) were selected (correct classification was 84.2%). For model 3, CCHL score (OR: 1.179, 95% CI: 1.013–1.371), height (OR: 1.087, 95%CI: 1.028–1.150), and CS-30 score (OR: 1.190, 95 % CI: 1.090–1.300) were selected (correct classification was 85.4%). For model 4, CCHL score (OR: 1.208, 95% CI: 1.041–1.402), height (OR: 1.068, 95% CI: 1.012–1.128), and TMT-A (OR: 0.985, 95% CI: 0.975–0.996) were selected (correct classification was 85.4%).

## 4. Discussion

This study aimed to explore factors associated with gait speed. Logistic regression analysis was performed with gait speed as the dependent variable and CCHL score, age, gender, height, physical function, and attentional function as independent variables. The results demonstrated that CCHL and height significantly affected gait speed in all models. Furthermore, analysis with CCHL as the only independent variable also found a significant impact on gait speed.

CCHL was developed by Ishikawa et al. [25] as a tool to evaluate public health literacy in the average citizen and comprises the higher-order interactive and critical literacies rather than functional literacy. CCHL defines interactive literacy as the competency and motivation to independently take action based on health knowledge with an understanding of health information, as well as participation in a healthy social group. Critical literacy refers to critically analyzing health information and being able to apply it on a more individual basis to daily events and situations and, when necessary, being able to utilize such information even when the surrounding environment is not supportive. In other words, those with a high CCHL score can be thought to have healthier habits and higher physical activity in all circumstances. A past study using CCHL [25] found that subjects with higher interactive and critical health literacies had healthier lifestyle habits (smoking, diet, exercise, etc.) and significantly fewer subjective symptoms. In a randomized controlled trial on improving health literacy in older adults, Uemura et al. [24] found that an active learning intervention improved health literacy, physical function (including gait speed), physical activity level, and cognitive function in the intervention group. Interestingly, no functional rehabilitation training was offered in the class, implying that the practice of healthy behaviors was a result of individual decisions and efforts made by subjects in their daily lives outside of the intervention. The present study also suggested that subjects with higher health literacy may have had healthier behaviors, such as striving to walk quickly during their physical activities. This is thought to be one reason why health literacy contributes to gait speed.

Gait speed is a product of step length and step count. Decreased gait speed in older adults is primarily due to a reduction in step length [42]. Height also has some level of relationship with step length and stride length [43], and this is likely why height was extracted as a factor influencing gait speed. CS-30 was extracted as a factor affecting gait speed, but knee extension strength was not. Burnfield et al. [44] explored the relationship between a decrease in normal gait speed and a decrease in lower limb strength in 81 community-dwelling older adult males and found that reduced knee extension strength and reduced normal gait speed were not significantly correlated. This is thought to explain why knee extension strength was not found to affect gait speed in the present study. However, that same study [44] found that reduced hip extension strength was significantly correlated with and was an independent predictor of reduced normal gait speed. Buckinx et al. [45] also reported a significant relationship between a decrease in normal gait speed and a decrease in hip extension strength. This may be the reason that the CS-30 score, which involves not only knee extension strength but also hip extension strength, was extracted as a factor influencing gait speed.

These findings clarify the association of health literacy with gait speed in community-dwelling older adults. Gait speed is a vital indicator of sarcopenia [46] and frailty [47,48] as well as a clinical indicator of disease status. Health literacy is comprised of individual factors such as age, education, and income; social and environmental factors including social support and health education; and interactive factors such as the systems for health and medical care and the public and patients.

## 5. Conclusions

This study was the first to reveal that health literacy is associated with gait speed, an essential indication for both health promotion and clinical purposes. Proactive health literacy initiatives will likely continue to become increasingly important in the community at the individual, country, and municipality levels as well as in health and medical care.

## 6. Limitations

The limitations of this study included the small sample size for logistic regression analysis, the highly disparate distribution of men and women, and the use of a cross-sectional design. Furthermore, this study focused on older adults participating in a community older adult social exchange program who did not require assistance. In other words, subjects were older adults who can be said to have relatively high health literacy. Older adults who do not or cannot participate in such programs are more likely to develop health and medical issues and should be studied in the future.

## Figures and Tables

**Figure 1 healthcare-08-00369-f001:**
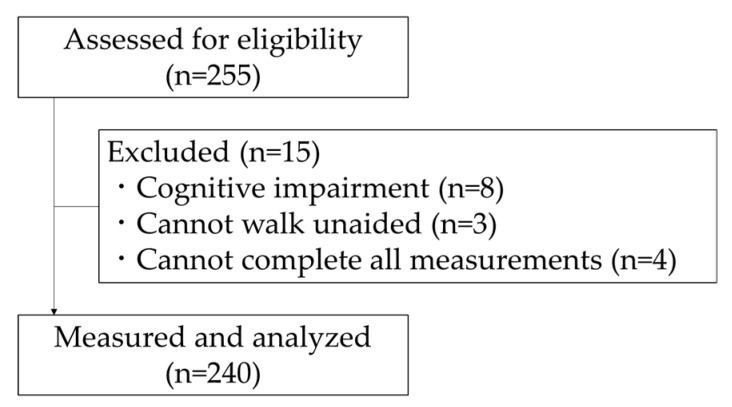
Flowchart of participation in the present study.

**Table 1 healthcare-08-00369-t001:** Subjects’ characteristics.

Variable	Total	Slow Gait Speed	Fast Gait Speed	*p*-Value	Effect Size
	(*n* = 240)	(*n* = 40)	(*n* = 200)		(φ, d)
Age (years)	73.8 ± 6.0	73.7 ± 5.2	73.8 ± 6.2	0.859	0.030
Gender					
male (n; %)	52 (21.7%)	8 (20.0%)	44 (22.0%)	0.779	0.018
female (n; %)	188 (78.3%)	32 (80.0%)	156 (78.0%)		
Education (years)	11.8 ± 2.3	11.9 ± 2.0	11.8 ± 2.3	0.849	0.030
Height (cm)	154.6 ± 7.7	151.0 ± 7.6	155.3 ± 7.6	0.001	0.560
Weight (kg)	53.3 ± 8.7	52.5 ± 8.3	53.4 ± 8.8	0.546	0.100
BMI (kg/cm^2^)	22.3 ± 3.0	23.0 ± 2.7	22.1 ± 3.1	0.114	0.270
Diseases					
Hypertension, yes (n; %)	94 (39.2%)	18 (45.0%)	76 (38.0%)	0.408	0.053
no (n; %)	146 (61.8%)	22 (55.0%)	124 (62.0%)		
Dyslipidemia, yes (n; %)	30 (12.5%)	7 (17.5%)	23 (11.5%)	0.295	0.068
no (n; %)	210 (87.5%)	33 (82.5%)	177 (88.5%)		
Diabetes, yes (n; %)	14 (5.8%)	3 (7.5%)	11 (5.5%)	0.709	0.032
no (n; %)	226 (94.2%)	37 (92.5%)	189 (94.5%)		
Cardiovascular disease,	18 (7.5%)	2 (5.0%)	16 (8.0%)	0.745	0.042
yes (n; %)					
no (n; %)	222 (92.5%)	38 (95.0%)	184 (92.0%)		
Respiratory disease,	7 (2.9%)	1 (2.5%)	6 (3.0%)	1.000	0.011
yes (n; %)					
no (n; %)	233 (97.1%)	39 (97.5%)	194 (97.0%)		
Osteoporosis, yes (n; %)	17 (7.1%)	2 (5.0%)	15 (7.5%)	0.745	0.036
no (n; %)	223 (92.9%)	38 (95.0%)	185 (92.5%)		
Orthopedic disease,	37 (15.4%)	4 (10.0%)	33 (16.5%)	0.299	0.067
yes (n; %)					
no (n; %)	203 (84.6%)	36 (90.0%)	167 (83.5%)		
CCHL (point)	19.3 ± 2.4	18.3 ± 2.7	19.5 ± 2.3	0.004	0.510
Grip strength (kg)	25.8 ± 6.5	23.6 ± 7.1	26.2 ± 6.4	0.020	0.410
Knee extension strength (%)	43.5 ± 10.7	39.5 ± 11.1	44.3 ± 10.5	0.009	0.460
Toe-grip strength (kg)	7.0 ± 3.0	5.8 ± 3.5	7.3 ± 2.9	0.005	0.490
Sit-up test (number)	9.0 ± 6.5	7.6 ± 6.1	9.2 ± 6.6	0.163	0.250
CS-30 (number)	21.1 ± 5.3	17.6 ± 5.1	21.8 ± 5.1	*p* < 0.001	0.820
Sit-and-reach test (cm)	34.9 ± 9.3	31.7 ± 10.8	35.6 ± 8.9	0.015	0.420
One-leg stance test (sec)	36.6 ± 36.0	17.0 ± 25.6	40.5 ± 36.5	*p* < 0.001	0.670
Gait speed (m/sec)	1.50 ± 0.21	1.16 ± 0.14	1.57 ± 0.15	*p* < 0.001	2.770
MMSE (point)	28.1 ± 1.8	27.9 ± 2.0	28.2 ± 1.8	0.397	0.150
TMT-A (sec)	105.5 ± 32.5	18.3 ± 35.3	101.8 ± 30.6	*p* < 0.001	0.720

Values are presented as means ± SD. Abbreviations: BMI, body mass index; CCHL, Communicative and Critical Health Literacy; CS-30, 30-s chair-stand test; MMSE, Mini Mental State Examination; TMT-A, Trail Making Test-A.

**Table 2 healthcare-08-00369-t002:** Logistic regression analysis with gait speed as the dependent variable.

Variable	OR	95%	*p*-Value
		Confidence Interval	
Model 1 *			
CCHL (point)	1.222	(1.062–1.405)	0.005
Model 2 ^†^			
CCHL (point)	1.231	(1.064–1.425)	0.005
Height (cm)	1.091	(1.033–1.152)	0.002
Model 3 ^‡^			
CCHL (point)	1.179	(1.013–1.371)	0.033
Height (cm)	1.087	(1.028–1.150)	0.003
CS-30 (number)	1.190	(1.090–1.300)	*p* < 0.001
Model 4 ^§^			
CCHL (point)	1.208	(1.041–1.402)	0.013
Height (cm)	1.068	(1.012–1.128)	0.017
TMT-A (sec)	0.985	(0.975–0.996)	0.006
Dependent variable: gait speed (fast or slow)			

Abbreviations: CCHL, Communicative and Critical Health Literacy; CS-30, 30-s chair-stand test; TMT-A, Trail Making Test part A; OR, Odds Ratio. * Adjusted for CCHL. ^†^ Adjusted for Model 1 plus Age and Gender, Height. ^‡^ Adjusted for Model 2 plus CS-30 and TMT-A. ^§^ Adjusted for Model 2 plus Knee extension strength and TMT-A.

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
