# Peer review of "The Association between Health Literacy and Gait Speed in Community-Dwelling Older Adults"

_healthcare, 2020, doi:10.3390/healthcare8040369_

Round 1
Reviewer 1 Report
Dear researchers,
Thank you very much for sending your modified article after the comments of the authors.
Reviewer 2 Report
Dear Authors,
Thank you for revising the manuscript as per suggestions. After revisions have been made, it makes the article more clarified and self-explanatory.
I, therefore, accept the manuscript without further recommendations for changes.
Thanks
Reviewer 3 Report
Accept in the present form. Congratulation!
This manuscript is a resubmission of an earlier submission. The following is a list of the peer review reports and author responses from that submission.
Round 1
Reviewer 1 Report
The authors provided a good presentation of study and the imperative of health literacy in health outcome is clearly represented. The conclusion of the study is of very high social/global importance and should be available for readers (in scientific and non-scientific fields) to study and apply in daily lives. However, I have following suggestions to make: 1. The introduction should end with a proper reason to perform the study and aim of the study. Thus the sentence, "there are limited number of studies....' in line 57 may be shifted to new paragraph should include sentence no. 63 and 64. this will improve the representation of introduction. 2. For results: although the tables are very clearly represented, the comparisons which are statistically significant should be highlighted for visual difference. To achieve this, the authors can format the statistical significant p-values to bold, italics, or assign asterisk (*) sign in superscript. Good luck!Author Response
Please see the attachment.

Reviewer 2 Report
The article presents an interesting study that requires some minor improvements to be published in Healthcare:
-Please include a figure or diagram showing the general experimental procedure used graphically.
-Explain in detail why women were not included in the study. Furthermore, it should be commented that future studies should include women and also comment on what to expect or consider in a study with women.
- Include a section that includes conclusions of the work.
Reviewer 3 Report
Dear Editors of Healthcare,
after carefully reading the manuscript you kindly provided for peer-reviewing, the following comments / issues are to be considered from my perspective:
Introduction:
Overall, the introduction is well written. However, it remains unclear for me why gait speed and health literacy should be connected in the first place. While the authors continue to explain the idea behind their work throughout the manuscript, at the end of the introduction this remains a question. I therefore would recommend that the authors explain the basic idea behind their research in more detail. Questions to be answered would be: Is there any evidence that health literacy does influence functional performance in older people? Can functional performance be enhanced by increasing health literacy?
Methods:
- The described number of participants is misleading. As is stated later, the actual number of participants is 240. As the 15 older persons not included were not dropouts but fulfilled an exclusion criteria, this should be reported accordingly.
- Please provide the defined inclusion criteria
- “education history” is unclear. From the analysis, it can be seen that education years were analyzed. However, that is not the same as education history. So please provide more detail on what you included in the data collection. If only education years are collected, you should at least differentiate between school years and professional education years.
- “History of chronic conditions”. Was just the list you state in brackets was reported or was any chronic condition was included? How did you defined chronicity, for example in orthopedic diseases?
- You should provide evidence for all measurement instruments that were used within the data collection of your study. Right now, none is presented on grip strength, knee extension strength, toe-grip strength and sit-and-reach.
- From your description, the difference between your measurements of sit—up-repetitions and the CS-30 is utterly unclear to me. Please provide a more detailed description on both measurements. You can include a description on how these two measurements are distinct to each other in the description.
- You performed the TMT-A, but not the TMT-B. As it is standard to perform both, you should provide reasons for excluding the TNMT-B
Results:
In table 2, you present group differences based on the differentiation between the fast walking and the slow walking group. Because you used the t-test for all these factors, I see a high probability for a type-I-error in your analysis. I would highly recommend of including a correction for this type of error, potentially a Bonferroni-correction.
Additionally, you should report your included independent variables in the methods section.
Discussion:
You conclude that health literacy influences gait speed (page 6, line 227/228). Furthermore, (line 231/232) you conclude that gait speed can be maintained by increasing levels of health literacy. I think this interpretation of your results is premature.
There are many factors contributing to slower gait speed in older people with high levels of evidence, while it is quite possible that health literacy is a contributor, too; but solely on the results of your study, the conclusion as you phrased it is not valid. We, in your study, can observe a correlation between gait speed and health literacy. Whether there is a direct line of reasoning between the two factors cannot be established by the results of your study, especially since you did not report a sample size calculation.
I would therefore recommend to reconsider your interpretation of your results and take into account the flaws of your study (which every study has), like lack of sample size, relatively low number of participants, highly disparate distribution of men and women, among others. Additionally, you should be able to establish a clear (albeit hypothetical) clinical consequence of your findings or a path to obtain such a path in orde to demonstrate the importance of your findings.
Reviewer 4 Report
Dear authors,
Thank you very much for sending your study to this magazine. The subject of your study is interesting. However, I think that the tables of results could be better organized, especially table 1. As for the discussion, it seems to me that it is very concrete and should explain in more detail the conclusions.
Regards.
